# Molecular Genetics of Acquired Temporal Lobe Epilepsy

**DOI:** 10.3390/biom14060669

**Published:** 2024-06-07

**Authors:** Anne-Marie Neumann, Stefan Britsch

**Affiliations:** Institute of Molecular and Cellular Anatomy, Ulm University, 89081 Ulm, Germany; anne-marie.neumann@uni-ulm.de

**Keywords:** mesial temporal lobe epilepsy, molecular genetics, transcription factors, pilocarpine, focal epilepsy, epileptogenesis

## Abstract

An epilepsy diagnosis reduces a patient’s quality of life tremendously, and it is a fate shared by over 50 million people worldwide. Temporal lobe epilepsy (TLE) is largely considered a nongenetic or acquired form of epilepsy that develops in consequence of neuronal trauma by injury, malformations, inflammation, or a prolonged (febrile) seizure. Although extensive research has been conducted to understand the process of epileptogenesis, a therapeutic approach to stop its manifestation or to reliably cure the disease has yet to be developed. In this review, we briefly summarize the current literature predominately based on data from excitotoxic rodent models on the cellular events proposed to drive epileptogenesis and thoroughly discuss the major molecular pathways involved, with a focus on neurogenesis-related processes and transcription factors. Furthermore, recent investigations emphasized the role of the genetic background for the acquisition of epilepsy, including variants of neurodevelopmental genes. Mutations in associated transcription factors may have the potential to innately increase the vulnerability of the hippocampus to develop epilepsy following an injury—an emerging perspective on the epileptogenic process in acquired forms of epilepsy.

## 1. Introduction

Epilepsy, one of the most common neurological diseases worldwide, is marked by recurrent seizures that emerge without any provocation (e.g., infection, trauma, drugs). It affects more than 50 million people of all ages [1]. Various forms of epilepsy, as well as several diseases that can be accompanied by epileptic seizures as a symptom (e.g., malnutrition), have been described. Epilepsies are most commonly categorized by seizure origin and pathogenesis (genetic vs. acquired). Temporal lobe epilepsy (TLE) is the most prevalent form of focal epilepsy, with seizure origins either in the mesial temporal lobe (e.g., the hippocampus) or in neocortical regions (lateral). Many cases of TLE are considered acquired; that is, they evolved as a consequence of a brain injury from perinatal or postnatal hypoxia, brain infections, or other traumatic injuries, as well as brain malformations and tumors, but a notable number of cases of TLE develop without a known cause [2]. Furthermore, complex febrile seizures are strongly associated with an increased risk to develop TLE: 30–50% of TLE patients have had febrile seizures in childhood [3]. However, the majority of children that have experienced febrile seizure do not develop TLE. It is not understood what leads to the development of epilepsy (“epileptogenesis”) in some patients compared to others.

The genetic background is estimated to be involved in the vast majority of epilepsy cases, either directly or indirectly [4]. Currently, 1215 and 354 genes are linked to “epilepsy” and “temporal lobe epilepsy”, respectively, in the disease gene net databank [5]. The majority of epilepsy-associated genes or variants are involved in intercellular communication or cellular structure, but subsets have also been linked to nucleic acid binding and transcriptional regulation [6,7,8]. Rodent models are a useful tool to elucidate the role these genes play in inherited or acquired epilepsies. Modeling methods include genetic manipulation, brain blood flow blockade, infections, kindling (repeated chemical or electrical stimulation of limbic brain regions), and *status epilepticus* (SE) models. The excitotoxic models share many similarities with human mesial TLE (MTLE; [9,10]). Using these models has greatly increased our understanding of epileptogenesis and how dysfunctional (molecular) genetics may be involved.

The manifestation of seizures in TLE is largely attributed to a permanent imbalance of inhibitory and excitatory signals, including the hippocampal circuits that lead to recurrent synchronized excess discharges of neurons. However, how this permanent imbalance is established and which role the genetic background plays is still not coherently understood. When trying to untangle epilepsy pathogenesis, we should try to distinguish among factors that influence (i) seizure onset (i.e., ictogenesis), (ii) the vulnerability to develop epilepsy following trauma (i.e., initiation of epileptogenesis), and (iii) the progression of epileptogenesis (i.e., chronification and worsening).

## 2. Hallmarks of Epileptogenesis

The development of acquired TLE is induced by an initial brain injury that starts a cascade of processes resulting in the restructuring of predominantly the hippocampus, amygdala, or lateral neocortex. This pathological process is referred to as epileptogenesis and frequently progresses after the first unprovoked seizure has occurred. Following a traumatic injury, a microenvironment is established that promotes seizures, neurodegeneration, and neuroinflammation, which often, but not always, further worsen the structural integrity [11,12]. Thanks to large epidemiological studies, we confidently know that various traumatic brain injuries increase the risk of developing TLE by lesioning crucial brain structures. However, how exactly these initial traumatic brain injuries trigger epileptogenesis remains rather elusive. This is especially apparent when considering the fact that neither of these potential causes is a definitive cause (e.g., [13,14]). Instead, multiple factors are involved, and individual traumatic thresholds must be reached to trigger epileptogenesis.

Epileptogenesis in MTLE alters the physiological hippocampal architecture. The histopathological hallmarks of an epileptic hippocampus besides neuroinflammation are (i) Ammon’s horn sclerosis, (ii) granule cell dispersion, and (iii) aberrant mossy fiber sprouting (for a more detailed review see, e.g., [15]). Neuronal loss happens predominantly in the CA1 and CA3 regions (cornu ammonis or Ammon’s horn), but inhibitory (GABAergic) interneurons and mossy cells in the hilar region, as well as some dentate gyrus granule cells, are also lost [16,17,18]. As is typical for neurodegenerative diseases, lost (neuronal) tissue is substituted by increased gliosis [19,20]. The loss of granule cells is one of the processes changing the tightly packed dentate gyrus layer. Additionally, newborn neurons show ectopic migration: instead of moving into the layer, cells remain in the subgranular zone or move into the hilar region (“granule cell dispersion”, [21,22]). Furthermore, the remaining granule cells have altered connectivity and functionality. Mossy fibers are the axons of these granule cells and project into the CA3 region. During epileptogenesis, mossy fibers sprout into the dentate gyrus, possibly trying to compensate for lost granule cells. Mossy fiber sprouting may strengthen hyperexcitability or counter it via inhibitory basket cells [23,24,25]; however, disease phenotype modulation (e.g., seizure frequency) does not consistently correlate with the amount of mossy fiber sprouting [21,26]. Though these hippocampal restructurings are found in most TLE patients and are a part of the epileptogenic process [9], it is not yet fully understood which are epileptogenic themselves and which are a response to recurrent seizures.

## 3. Disease-Associated Cellular Events

To investigate the underlying mechanisms involved in epileptogenesis, several “-omics”-based studies have been performed (e.g., [27,28,29,30]). Rodent animal studies are typically conducted to involve the “latent” phase, that is, the phase before the first unprovoked seizure, which refers to a timeframe of up to a week post-injury depending on the model, species, and strain. One has to note that it is extremely difficult in such studies to distinguish among effects being a consequence of the brain injury, initiators of epileptogenesis, or processes further along the disease progression, especially without invasive EEG monitoring. Corresponding data from human samples are not sufficiently available; samples are usually taken during surgery after failed pharmacological treatment; therefore, they often represent a biased sample from late disease progression. Overall, these studies have commonly found a deregulation of neuroinflammation, neurogenesis, neurodegeneration, and plasticity-associated signaling [31]. However, the majority of investigations lack spatial or single cell type-resolution and, consequently, offer only a limited perspective.

### 3.1. Neuroinflammation and Astrocytes

Neurons, microglia, and astrocytes react to a brain injury by the secretion of proinflammatory signals. This early activation mostly occurs in response to brain injury, but a proinflammatory environment can enhance the excitability of neurons and propagate epileptogenic seizure circuits [32]. Besides this early immune response, a TLE-triggering SE, as well as the proinflammatory environment during epileptogenesis, is able to weaken the blood–brain barrier, which opens the door for peripheral immune cells and cytokines to enter the brain. It was shown that T-cells in both the pilocarpine and kainate model migrate into the hippocampal parenchyma [33,34], which might enforce chronification of the disease. By regulating cytokine production (e.g., IL-1β, IL-18, IL-6, and TNF-α), attenuation of hippocampal damage and loss of neurons during epileptogenesis is possible [35,36]. Overexpression of the synaptopathy gene *ADAM10* (A disintegrin and metalloproteinase domain-containing protein 10) was shown to be neuroprotective for cell death from SE-induced epilepsy, whereas a dysfunctional protein increased neuronal damage—both via the regulation of neuroinflammation and proinflammatory signaling [37,38].

Neuroinflammation is triggered by proinflammatory signals from neurons, microglia, and astrocytes. Interestingly, astrogliosis is a robust feature in all stages of epileptogenesis and is found across different times in animal models and humans [39,40]. Reactive astrocytes facilitate a proinflammatory state as a response to brain injury and lose their neuroprotective role (i.e., control of the blood–brain barrier and extracellular homeostasis; [41]). Genetically induced chronic reactive astrogliosis by itself or as a characteristic feature of a mutation is able to generate seizures in mice [42,43]. Moreover, a study of brain samples from healthy humans and patients with epilepsy showed that while neurogenesis declined with the duration of MTLE, immature astrocytes are consistently increased [40]. This hints toward a role of newborn astrocytes and not newborn neurons in disease progression.

### 3.2. Network Connectivity

Epileptogenesis is characterized by the remodeling of neuronal plasticity and connectivity which creates hyperexcitable neurons and positive feedback loops. Though only a minor population of epilepsy patients can be described by inherited or de novo monogenic mutations, the relatively small subset of known pathogenic variants is mostly associated with ion channels, neurotransmitter receptors, neuronal metabolism, and synaptic complexes [6,44]. These genes are obviously fundamental in overall brain functionality, and therefore, these forms of epilepsy often present with a diverse set of accompanying neurological symptoms. Nevertheless, this highlights how basic mechanisms of neurotransmission are involved in epilepsy pathogenesis (for a more detailed review, see [45]).

A TLE-triggering trauma and frequent epileptic seizures induce neuronal death. This happens within hours after the initial brain injury and continues for several days in animal models [46,47,48]. Predominantly, principal cells of the dentate gyrus (i.e., granule cells) and within the CA1 region (i.e., pyramidal neurons) are lost. Another fraction of lost cells are GABAergic inhibitory interneurons [49,50,51]. Interneurons tightly control the excitatory signaling and information processing within the hippocampal circuitry. The selective ablation of interneurons does induce epileptic seizures and epileptogenesis in mouse models [52,53]. In the pilocarpine mouse model of TLE, the loss of GABAergic interneurons in the dentate gyrus was correlated with seizure frequency but not with the extent of mossy fiber sprouting or astrogliosis [54]. Confirming this finding, the reprogramming of reactive glia into interneurons was able to reduce chronic seizure activity in TLE models [55,56].

The loss of interneurons or a functionally specific subset of interneurons seems to be a crucial part of epilepsy pathogenesis. Ictal onset sites correlate with granule cell-associated GABAergic interneuron loss [57]. The selective silencing of parvalbumin (PV) or somatostatin (STT) interneurons, subtypes specifically abundant in the dentate gyrus, was able to induce spontaneous epileptic seizures [58,59]. The genetic depletion of an anti-epileptic pathway (NRG1/ErbB4; neuregulin 1, receptor tyrosine–protein kinase erbB-4) in forebrain PV interneurons led to an increase in epileptogenic mossy fiber sprouting and spontaneous seizures [60]. Moreover, the interneurons that survive may further propagate epileptogenesis by fulfilling more of an excitatory and seizure-synchronizing role. Evidence indicates that, despite an overall loss of interneurons, GABAergic synapses on principal cells increase [61]. Those newly established connections are likely dysfunctional or follow an overall excitatory route [50,61]. GABAergic dentate interneurons appear to become hyperactive initially and again later in epileptogenesis [62]. Early in epileptogenesis, PV and STT interneurons are differently affected [63] and overall interneuron network desynchronization was proposed as a mechanism involved during disease progression [64,65]. Additionally, the relationship between interneurons and the activating hilar mossy cells may be a key regulator of granule cell activity [18,66]. Taken together, interneurons are fundamentally involved in epileptogenesis. Innate dysfunction of interneurons can induce epilepsy in mice; deregulated activity may be a predisposing factor in humans.

The loss of principal cells and interneurons is accompanied by a presumably compensatory increase in neurogenesis processes. Indeed, adult-born neurons appear to have neuroprotective functions after an epileptogenic brain injury and for initial seizure generation [67,68]. In such studies, neurogenesis was manipulated for prolonged time before the injury. In contrast, nestin-mediated inhibition of adult neurogenesis before SE proved to attenuate epileptogenesis [69]. The role of adult neurogenesis is complex, and its influence on epileptogenesis seems to depend on the experimental setup [70]. After an epileptogenic injury, newborn neurons show morphological diversity in both rodents and humans: while some integrate apparently normally into the dentate gyrus layer, others ectopically migrate and show pathological activity and connections [71,72]. Labeling newborn granule cells before and after a TLE-triggering SE showed that neurons born before the SE and those born after SE both contribute to mossy fiber sprouting, however, only the latter migrated ectopically [73]. Silencing the activity of neurons born just prior to a TLE-triggering SE (i.e., 2 days) for two weeks post-SE reduced spontaneous seizures and ectopic hilar granule cells. This rescued some physiological projections but did not reduce mossy fiber sprouting [21]. It was proposed that these ectopic granule cells act as hub cells, which facilitate the breakdown of the gating of the dentate gyrus. Interestingly, this aberrant neurogenesis appears to be largely regulated by GABA, which further supports the idea of miswired interneurons contributing to hyper-excitatory feedback loops [21].

## 4. Molecular Regulation of Epileptogenesis

The analysis of human hippocampal transcriptomes led to a vast increase in the number of genes considered to be involved in the epilepsy pathogenesis. For example, transcriptome analysis of brain tissue from patients with MTLE and hippocampal sclerosis showed an enrichment of deregulated genes related to neuron development, differentiation, and synaptic signaling, as well as a network defined by hub genes involved in CNS development, axon guidance, and neuronal differentiation, including two transcription factors [29]. Furthermore, Zhang et al. found hundreds of differently methylated genes between TLE patients with and without hippocampal sclerosis [30]. Such investigations offer great insight into modulated pathways underlying the progression of epilepsy. For the initiation of epileptogenesis, however, transcriptomics studies of rodent brain tissue after epileptogenic trauma but before the first unprovoked seizures are crucial (e.g., [28,74]). It is tempting to assume that certain major regulating pathways are activated following trauma, which may initiate processes involved in hippocampal restructuring. Furthermore, an innate imbalance in the activity of these pathways could be a risk factor for acquired epilepsy.

### 4.1. Major Pathways

The mTOR pathway has long been a target of antiepileptic research due to genetic forms of epilepsy with mutations within the mTOR pathway (mammalian target of rapamycin; e.g., tuberous sclerosis 1/2 (TSC1/2) and phosphatase and tensin homolog (PTEN); [75,76]). However, results from the therapeutic use of mTOR inhibitors have been rather controversial in acquired epilepsy [77,78]; the cell-type specific manipulation of mTOR signaling may prove more successful [78,79]. Inducing a hyperactive mTOR by the deletion of *Pten* in adult-born granule cells is sufficient to cause epileptogenesis [80,81]. Of note, already very few of these aberrant adult-born granule cells led to histopathological features of epilepsy and seizures [81]. Other promising targets are the neurotrophic BDNF (brain-derived neurotrophic factor) and its receptor TrkB (tropomyosin receptor kinase B). BDNF is a key regulator of adult neurogenesis and synaptic transmission. BDNF/TrkB signaling is enhanced in patients with TLE, and ablation or inhibition of the pathway in experimental models appears to suppress epileptogenesis [82,83,84]. Furthermore, systematic inhibition of TrkB heavily increased cellular death in the CA1/3 region one day following SE. Thus, at least initially, TrkB acts neuroprotectively [85]. Consequently, targeting the epileptogenic vs. the neuroprotective path of TrkB was successfully tested [85]. Different cells in the hippocampus may follow either of these paths. Although the genetic deletion of TrkB in neurons or astrocytes proved to be neuroprotective following SE, astrocytic TrkB may be a more promising regulator of cell death and epileptogenesis [86,87]. Furthermore, anticonvulsant interneurons seemingly require TrkB for their suppressive function [88]. The Wnt/ß-catenin (wingless and Int-1) pathway is a major signal transduction pathway of cell proliferation and differentiation. Wnt increases in neurons early in epileptogenesis and Wnt/ß-catenin downregulation, or dysfunction appears to be neuroprotective [89,90]. Signaling of mTOR was associated with these effects [90].

Overall, major regulating pathways are heavily involved in epileptogenesis. Due to cell-type specific regulation and functions, it appears that potentially beneficial effects in one cell may be canceled out by the detrimental effects in another. Additionally, effects in peripheral tissues need to be considered when trying to pharmacologically target major regulators. Rather, a more nuanced approach could increase understanding and the therapeutic success rate. For example, the activation of reelin, a regulator specifically of neuronal migration and differentiation, is anti-epileptogenic [91,92]. Moreover, genetic variants of reelin are associated with epilepsy and TLE development [93,94]. Neurons, microglia, and astrocytes react to a brain injury with the secretion of proinflammatory signals. This early activation mostly occurs in response to the brain injury, but a proinflammatory environment can enhance the excitability of neurons and propagate epileptogenesis.

### 4.2. Role of Transcription Factors

Transcription factors are often hub genes; they regulate gene expression networks. As such, they are involved in starting and regulating major inflammatory, metabolic, or proliferatory pathways. For example, during cell differentiation, both postnatal and in the adult brain, transcription factors are expressed in a specific spatiotemporal and combinatorial manner to determine cellular fate and connectivity [95,96,97]. Transcription factors offer the opportunity to modulate functions of specific cells at specific timepoints during epileptogenesis—potentially starting immediately following an injury.

Prophylactic treatments trying to attenuate brain trauma and neuroinflammation to block epileptogenesis in mice lead to a downregulation of activity-dependent immediate early gene (IEG) transcription factors like c-Fos or Jun [98,99]. The serum response factor SRF, which regulates IEG responses to traumatic injury, is upregulated post-SE and adult neuronal ablation of *Srf* in mice highlighted a dual role of the transcription factor in epileptogenesis: an increased resistance to induce a TLE-triggering SE but also an increased frequency of spontaneous seizures post-SE [74,100]. Another activity-dependent transcription factor is CREB (cAMP response element-binding protein). Its associated gene expression is upregulated in granule cells and inhibitory GABAergic interneurons 4 h after SE. Later in epileptogenesis, expression is shifted more toward an increase in microglia and astrocytes [101,102]. Constitutionally or conditionally decreasing CREB proved to be suppressive of epileptogenesis [101,103]. However, long-term inhibition of CREB could also trigger neurodegeneration [104].

Cell death in the CA1/3 region of the hippocampus is a pronounced feature of TLE. The expression of apoptotic tumor suppressor protein p53 is upregulated in MTLE patients and ablating its negative regulator *Chop* (DNA damage-inducible transcript 3 or C/EBP homologous protein) increased cellular death [105,106]. The proinflammatory signaling following brain trauma and cell death appears to be a major subsequent step in initiating epileptogenic changes in the hippocampal connectome. The transcription factors Nrf2 (nuclear factor erythroid 2-related factor 2) and NFkB (nuclear factor kappa-light-chain-enhancer of activated B cells) are key regulators of oxidative stress and neuroinflammation. Several studies have shown an increase in NFkB and associated target cytokines following SE, which occurs in a temporal order: within a few hours (i.e., 4 h) in neurons and days later in astrocytes and microglia [107]. Moreover, antagonizing NFkB has been shown to modulate disease onset, cell death, and neuroinflammation [107]. Notably, NFkB inhibition also enhanced susceptibility to kainate-induced seizures [108]. This hints at a neuroprotective role of presumably neuronal NFkB signaling during trauma before the onset of epileptogenesis. NRF2 and its target genes are increased in the TLE of humans and mice, with a peak at about 72 h in experimental animal models [109,110]. It is predominantly upregulated in the neurons of the CA1/3 region and the astrocytes of CA1 post-SE [110]. Overexpression or activation of NRF2 showed a neuroprotective potential of the transcription factor [109,111]. Hippocampal sclerotic TLE patients with seizure recurrence post-surgery were found to have a dysfunctional antioxidative response in parallel with a reduced *NRF2* expression [112]. Furthermore, proposed antioxidative treatments in epilepsy treatment seem to work via the upregulation of NRF2 (e.g., [91,113,114]). The antioxidative ketogenic diet seems to modulate both NFkB and NRF2 to attenuate epileptogenesis [115]. In addition to the aforementioned well-known inflammatory regulators, other transcription factors have been shown to modulate epileptogenesis via neuroinflammation. The activation of the circadian transcription factor Rev-Erbɑ (reverse strand of ERBA, also known as nuclear receptor subfamily 1 group D member 1) during the first week of epileptogenesis, led to reductions of astrocytosis, microgliosis, and neuronal death [36]. The silencing of *Notch-1* (neurogenic locus notch homolog protein 1) increased neuroinflammation and neuronal death [116]. Furthermore, the proinflammatory and CREB-associated activating transforming factor 3 (ATF3) was upregulated in the hippocampus of MTLE patients [117]. Modulating primarily neuroinflammatory transcription factors shortly following TLE-inducting brain trauma was one of the most consistent recent treatment proposals. Clearly distinguishing the role these factors play in each involved cell type during initial trauma and early epileptogenesis will help refine treatments and potentially increase the understanding of genetic risk factors.

Transcription factors modulate the expression of neurotransmitter receptors and ion channels, thus changing neuronal plasticity and excitability. An increase in MTF1 (metal regulatory transcription factor 1) was associated with the increased excitability of neurons in acquired epilepsy by the altered expression of channels (channelopathies; [118]). Similar, REST/NRSF (repressor element 1 silencing transcription factor/neuron-restrictive silencer factor) is a regulator of channel expression and overexpressed in experimental models and humans with MTLE [119,120]. In a kindling model of TLE, resistant animals had higher protein levels of REST/NRSF, which may highlight a neuroprotective role against ictogenesis and hyperexcitability [121,122]. Furthermore, the downregulation of the BMAL1-PCDH19 (brain and muscle ARNT-Like 1, protocadherin 19) axis seems to be involved in the hyperexcitability of neurons following trauma and may explain daytime-dependent variations in seizure frequency [123].

More recently, the focus of antiepileptogenic research has shifted toward increasing healthy neurogenesis or blocking the generation and activity of aberrant neurons. Neurogenesis, which is involved in histopathological epilepsy hallmarks, like ectopic granule cell migration and mossy fiber sprouting, is tightly organized by the sequential expression of transcription factors. Transcription factors active in neurodevelopment are operative during epileptogenesis, thus possibly modifying epileptogenesis [124,125]. For example, MEF2A (myocyte-specific enhancer factor 2A), a regulator of synaptogenesis, was downregulated in the cortical neurons of TLE patients [126]. Differentially expressed microRNAs found in the human hippocampus were postulated to regulate mRNA expression of SOX11—a transcription factor of neuronal maturation [127]. Regulators of transcription involved in neurodevelopment are increasingly associated with epileptic encephalopathies and neurodevelopmental disorders with seizures, and therefore, could potentially also be modulators of acquired TLE (e.g., CHD2 [128], SETD5 [129,130], SOX11 [131], and BAF complex subunits [132,133]). Among the BAF complex subunits are the BCL11 transcription factors (B-cell lymphoma/leukemia 11). Some children with a diminished expression of the transcription factor BCL11B exhibit epileptic seizures [134]. This transcription factor modulates neurogenesis, mossy fiber sprouting, and synaptic function during development and in adults [135,136,137,138], and therefore, may also play a role in TLE. Additionally, BCL11B is highly expressed in GABAergic interneurons of target brain regions, but its functions here are not yet understood [139,140]. As mentioned before, interneurons play a fundamental role in epileptogenesis, and therefore, an innate dysfunction based on the altered activity of such a transcription factor may promote epilepsy development. Genetic variations of its paralog, BCL11A, have also been increasingly linked to epilepsy syndromes, including TLE [141,142,143]. Lastly, overexpression of the transcription factors NEUROD1 (neurogenic differentiation 1), ASCL1 (achaete-scute homolog 1), and DLX2 (distal-less 2) in reactive glia allowed for a cell fate switch from (reactive) astrocytes to interneurons, which reduced seizure frequency and neuronal loss in TLE models [55,56].

Many transcription factors follow a very specific spatiotemporal expression. Their function in organizing neurodevelopment and neuronal/synaptic maintenance make them prime candidates to modulate epileptogenesis. Besides their potential as therapeutic targets during epilepsy development, transcription factors may be innately deregulated and subsequently predispose people to epileptogenesis. This notion is reinforced by several transcription factor-associated epileptic encephalopathies. However, mechanistic knowledge in the context of post-traumatic epilepsy is lacking.

## 5. Genetic Predisposition

For a long time, research on TLE has been focused on finding novel therapeutic targets to modulate epileptogenesis as effectively as possible or to inhibit ictogenesis. With the rise of genetic sequencing methods and large-scale epidemiological studies, (poly-)genetic risk factors for most diseases have emerged. Thus, more and more studies indicate that a patient’s genetic background may be involved in their susceptibility to develop epilepsy. Family history of epilepsy, notably that of the mother, is associated with an increased risk of developing epilepsy in early life [144,145,146,147]. Mendelian inheritance does not sufficiently explain the majority of familial accumulated epilepsies, and interactions of rare and common variants are likely [2,148]. The majority of epileptic patients do not have a known affected relative or identified pathogenic genetic variants. This suggests a complex interaction of genetic background and environmental factors in the development of epilepsy syndromes in general [149]. Understanding such genetic predisposition in the case of acquired TLE offers the potential to intervene during the initial injury and before epileptogenesis has manifested.

MTLE was largely considered a purely traumatically-induced form of epilepsy. This perspective is increasingly challenged. In a small study from 2018, Cvetkovska et al. found 17% of MTLE patients had a family member (varying from first degree to third degree) with a form of epilepsy [150]. Of these, in at least two families, an interplay of genetic predisposition and traumatic brain injury or febrile seizures was observed. Furthermore, a Danish population-based study evaluated risk factors for the development of epilepsy following traumatic brain injury in children and young adults. A positive family history of epilepsy increased the relative risk after mild, severe, and repeated brain injury [144,146]. A recent study on 105 MTLE patients found eleven pathogenic somatic variants enriched in the hippocampus, of which the majority was predicted to activate a key proliferative pathway [151]. Together, these data strongly suggest that the vulnerability to acquire MTLE can be modified by the genetic landscape. Notably, studies on wildtype inbreed rodent models confirm a varying vulnerability to SE and epilepsy-induction based on the genetic background [152,153]. More elaborate sequencing methods have given rise to an abundance of genetic variants that may contribute to epilepsy risk in humans. However, many of such studies fail to be replicated, and the genetic contribution likely depends on the injury inducing TLE [154].

Variations of genes that are associated with mono- or polygenetic epilepsies have the potential to predispose people to TLE, too. Single-nucleotide polymorphisms (SNPs) within the epilepsy gene *SCN1A*, a subunit of a voltage-gated sodium channel, were associated with the risk of developing TLE or febrile seizures to varying degrees [155,156]. Mutations of the reelin *RELN* gene are associated with an autosomal familial form of epilepsy; however, other variations were found in patients with sporadic MTLE without any family history [93,94]. Both have an important role in the development and maintenance of the hippocampus [157,158]. Genetic variants related to ion channels (e.g., SCN1A), calcium signaling (e.g., CALHM1), or neurotransmitter actions (e.g., GABBR1, 5-HTT) have all been linked to common epilepsies, including TLE [7]. Similarly, SNPs of key neuroregulatory genes were associated with TLE, including known epilepsy-associated genes *NRG1*, *BDNF*, or the *PDYN* promoter (prodynorphin; [159,160,161,162]). Furthermore, psychiatric comorbidities increase the risk of acquiring epilepsy, suggesting common dysregulated molecular pathways involved in the pathogenesis of both [146]. Specific polymorphisms of schizophrenia-related gene *DTNBP1* (dystrobrevin-binding protein 1) were found to either increase or reduce the risk for TLE [163]. The apolipoprotein E ε4 genotype seems to be associated with an increased risk of MTLE [164,165]. These studies have strengthened the idea that genetic variations in epileptogenesis-associated genes and pathways can act as risk factors for the development of acquired TLE.

Inflammatory and neurodevelopmental transcription factors play a pronounced role in epileptogenesis. Consequently, genetic variants of some of the previously mentioned transcription factors (see Section 4.2) were found to increase TLE or general epilepsy risk. SNPs in *NRF2* were identified as risk factors for TLE and drug resistance [166]. A likely pathogenic variant of CHD2 (chromodomain helicase DNA binding protein 2) was found in a case of adult-onset TLE, without mention of prior brain injury [167]. However, as of now, genetic variations of such prominent transcription factors or closely associated molecules are rather linked to severe neurodevelopmental disorders accompanied by epileptic encephalopathy than a genetic risk for acquired TLE. For example, missense mutations of the CREB binding protein induces Rubinstein–Taybi syndrome or a related but milder phenotype [168,169]. Mutations in the BAF subunit ACTL6B (actin-like protein 6B) cause neurodevelopmental delay and epilepsy encephalopathy likely via a loss of dendrites and an increase in neuronal hyperexcitability [132,170]. Similarly, pathogenic variants of BAF subunits Bcl11a und b induce intellectual disability disorders often associated with epileptic syndromes [134,141,143,171,172]. However, the first evidence indicates that mildly pathogenic variants of these genes could increase overall epilepsy risk. A common variant of *BCL11A* has recently emerged as a general risk factor for epilepsy and drug resistance [142]. This highlights the potential role specific transcriptional–translational regulation could play not only in epileptogenesis but in establishing a favorable baseline microenvironment for post-traumatic epilepsy induction.

It is becoming more and more apparent that TLE etiologies are far more complex than originally thought and often require multiple hits (genetic and traumatic) to manifest. To specifically study potentially associated genes, new mutant animal models are needed to fully grasp their role in post-traumatic early epileptogenesis or disease onset. Expanding research on the genetic predisposition of TLE offers the chance to find predictors for the epileptogenic risk following a lesion or febrile seizures in children or even before any injury has occurred and allows for the close monitoring of patients at risk.

## 6. Conclusions

The chronology of post-traumatic epileptogenesis starts with stress in the brain parenchyma, which leads to neuronal death, neuroinflammation, astrocytosis, delayed apoptosis of regulatory neurons (e.g., interneurons, mossy cells), compensatory neurogenesis, and altered connectivity. This establishes an hyperexcitable environment and a vicious cycle with consistently stressed neurons by unprovoked recurrent seizures. Despite extensive knowledge about the changes in neuronal plasticity and excitability during epileptogenesis, refractory epilepsy and seizure control remains a problem for a large portion of patients with MTLE [1,45]. Pharmacological manipulation of key epileptogenic cellular events by the regulation of major pathway molecules has led to mostly controversial results. This is largely due to a poor understanding of the master epileptogenic process, which could prevent disease onset when manipulated. Research focusing more deeply on cell type-specific functions in the hippocampus indicates a strong context dependence for many molecular pathways (Table 1). Novel technologies of spatial transcriptome analyses could paint a detailed picture of expression patterns across epileptogenesis [173]. The regulation of transcription via transcription factors or non-coding RNAs may offer more specific targeting of specific cells at specific points during differentiation after trauma [174]. Given that aberrant newborn granule cells are likely substantial drivers of epileptogenesis [21,81], transcription factors of adult neurogenesis may be a key to prevent or avert TLE development. Besides actively modulating epileptogenesis, acquired or inherited variations of such transcription factors may alter susceptibility of the brain to the epilepsy-inducing mechanisms of an injury. Evidence hints toward potentially underlying innate deregulations involved in the pathogenesis of acquired epilepsies. Similarly, mutations in potential epilepsy risk genes, like *BCL11A/B*, leading to neurodevelopmental delay and intellectual disability syndrome, are often but not always accompanied by epileptic seizures [134,141,142,143]. Combining different experimental animal models of TLE with genetic manipulation and large-scale sequencing studies in well-characterized populations with acquired TLE will further elicit the interplay between genetic background and traumatic injury in disease manifestation. Understanding these interactions will advance therapeutic options and prophylactic monitoring.

## Figures and Tables

**Table 1 biomolecules-14-00669-t001:** Selection of candidate targets for (genetic) manipulation of epileptogenesis.

Mechanism	Target	Type	Specific Manipulation	Effect	Reference
*Neuroinflammation*	TNF-α	Cytokine	Knockout microglial TNFa	No Effect	Henning et al., 2023 [35]
	Cdk5	Endocrine factor	Endothelial Cdk5 ablation	Astrocytosis, seizure induction	Liu et al., 2020 [43]
	mTOR	Protein kinase	mTOR-deficient microglia	Increased neuronal loss, more severe seizures	Zhao. et al., 2020 [78]
	BDNF	Neurotrophic factor	Astrocytic overexpression	Increased seizure duration	Fernándes-García et al., 2020 [86]
	TrkB	Neurotrophic receptor	Astrocytic deletion	Decreased neuronal loss	Fernándes-García et al., 2020 [86]
*Interneurons*	Ascl1, Dlx2	Transcription factors	Retrovirus-driven expression in reactive glia	Reprogramming to interneurons, seizure reduction	Lentini et al., 2021 [55]
	NeuroD1	Transcription factor	Overexpression in reactive glia	Reprogramming to interneurons, seizure reduction	Zheng et al., 2022 [56]
	PV	Calcium-binding protein	Silencing of interneuronal subset	Seizure induction	Drexel et al., 2017 [58]
	STT	Inhibitory growth factor	Silencing of interneuronal subset	Seizure induction	Drexel et al., 2022 [59]
	ErbB4	Receptor tyrosine kinase	Depletion of ErbB4 in PV-interneurons	Increased seizure frequency and mossy fiber sprouting	Tan et al., 2011 [60]
	CST	Neuropeptide	Ablation of interneuronal subset	Seizure induction	Hill et al., 2019 [88]
	TrkB	Neurotrophic receptor	Deletion in cortistatin-expressing neurons	Seizure induction	Hill et al., 2019 [88]
*Principle neurons*	Nestin	Intermediate filament	Ablation of nestin-expressing cells (=cells in development)	Ablation of neurogenesis, reduced seizure frequency	Cho et al., 2015 [69]
			Retrovirus-driven silencing of newborn neurons	Reduced seizure frequency and ectopic neurons	Lybrand et al., 2022 [21]
	PTEN	Tumor suppressor	Tamoxifen-Gli1-driven deletion of PTEN	Hyperactive mTOR in adult-born granule cells, seizure induction	Pun et al., 2012 [81]
	PTEN	Tumor suppressor	Tamoxifen-Gli1-driven deletion of PTEN to variable degrees	Hyperactive mTOR in adult-born granule cells, seizure induction	LaSarge et al., 2021 [80]
	EphB3 *	Receptor tyrosine kinase	Exogenous stimulation	Fewer neuronal progenitors, reelin increased, clinical manifestation attenuated	Liu et al., 2018 [91]
	SRF	Transcription factor	Adult neuronal ablation in forebrain	Increased seizure frequency	Losing et al., 2017 [74]
	Bmal1	Circadian transcription factor	Syn1-adeno-associated virus-driven knockout in granule cells	Increased susceptibility to seizure induction	Wu et al., 2021 [123]
*Unspecific*	CREB	Transcription factor	Constitutive knockdown	Clinical manifestation attenuated	Zhu et al., 2012 [101]
	CREB	Transcription factor	Inducible repression	Clinical manifestation attenuated	Zhu et al., 2015 [103]
	RevErb-a *	Circadian transcription factor	Agonist SR9009	Reduction in neuronal damage and neuroinflammation	Yue et al., 2020 [36]
	p53	Tumor suppressor	Ablation of negative regulator	Increased neuronal loss, increased seizure frequency	Engel et al., 2013 [105]
	Nrf2	Transcription factor	Adeno-associated virus-driven overexpression	Reduced seizure frequency, less neuroinflammation	Mazzuferi et al., 2013 [109]
	Nrf2 *	Transcription factor	Activation by RTA 408 via KEAP1	Reduced seizure frequency, less neuronal damage	Shekh-Ahmad et al., 2018 [111]
	Notch-1	Transmembrane receptor	Downregulation by silencing of miRNA-146a	Decreased neuronal loss	Huang et al., 2019 [116]

* Highlights nongenetic manipulation.

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
