# Peer review of "Molecular Genetics of Acquired Temporal Lobe Epilepsy"

_biomolecules, 2024, doi:10.3390/biom14060669_

Round 1

Reviewer 1 Report

Comments and Suggestions for Authors

The authors present a comprehensive review of the cellular and molecular mechanisms, as well as the possible genetic predisposition, associated with acquired temporal lobe epilepsy (TLE).

After an introduction to the subject of the review, the authors describe the histopathological hallmarks of epilepsy, concluding that while these alterations are related to epilepsy, it is not clear whether they are a cause or a consequence of recurrent seizures.

The authors then summarize the cellular events linked to epileptogenesis, such as neuroinflammation and the remodelling of neuronal plasticity and connectivity. Our current understanding of these processes is largely based on studies performed in rodent models.

The review also thoroughly examines the major molecular pathways potentially involved in epileptogenesis and the role of certain transcription factors and their modulation in the development of epilepsy.

Finally, the authors describe the potential genetic predisposition that may render certain individuals susceptible to epilepsy following trauma. They discuss the genes in which SNPs associated with this predisposition have been identified, highlighting the complexity of genetic regulation and the susceptibility to developing acquired TLE.

The review underscores the complexity of identifying potential treatments related to the regulation of cellular and molecular mechanisms. In some cases, regulation differs across various cell types and can be beneficial or detrimental depending on the cell involved. Furthermore, the modulation of certain transcription factors might have beneficial effects in the short term but adverse effects in the long term.

Additionally, I noticed a small error: the title of section 5, "Genetic Predisposition," is repeated. The second section with this title seems to be the conclusion of the article.

Overall, I find this review to be highly informative and relevant for researchers working in the field of epilepsy. It provides valuable insights and a detailed synthesis of current knowledge, making it a worthwhile contribution to the literature. Therefore, I recommend its publication.

Author Response

Response to Reviewer 1 Comments

1. Summary

We would like to thank you very much for taking the time to review our manuscript. Please find the corresponding corrections highlighted and in track changes in the re-submitted files.

2. Questions for General Evaluation

Reviewer’s Evaluation

Response and Revisions

Is the work a significant contribution to the field?

Are there appropriate and adequate references to related and previous work?   

3. Point-by-point response to Comments and Suggestions for Authors

Comments:

The authors present a comprehensive review of the cellular and molecular mechanisms, as well as the possible genetic predisposition, associated with acquired temporal lobe epilepsy (TLE).

After an introduction to the subject of the review, the authors describe the histopathological hallmarks of epilepsy, concluding that while these alterations are related to epilepsy, it is not clear whether they are a cause or a consequence of recurrent seizures.

The authors then summarize the cellular events linked to epileptogenesis, such as neuroinflammation and the remodelling of neuronal plasticity and connectivity. Our current understanding of these processes is largely based on studies performed in rodent models.

The review also thoroughly examines the major molecular pathways potentially involved in epileptogenesis and the role of certain transcription factors and their modulation in the development of epilepsy.

Finally, the authors describe the potential genetic predisposition that may render certain individuals susceptible to epilepsy following trauma. They discuss the genes in which SNPs associated with this predisposition have been identified, highlighting the complexity of genetic regulation and the susceptibility to developing acquired TLE.

The review underscores the complexity of identifying potential treatments related to the regulation of cellular and molecular mechanisms. In some cases, regulation differs across various cell types and can be beneficial or detrimental depending on the cell involved. Furthermore, the modulation of certain transcription factors might have beneficial effects in the short term but adverse effects in the long term.

Additionally, I noticed a small error: the title of section 5, "Genetic Predisposition," is repeated. The second section with this title seems to be the conclusion of the article.

Overall, I find this review to be highly informative and relevant for researchers working in the field of epilepsy. It provides valuable insights and a detailed synthesis of current knowledge, making it a worthwhile contribution to the literature. Therefore, I recommend its publication.

Response 1: Thank you for your positive evaluation of our work. As requested by reviewer 1, we have corrected the error and changed the title of section 6 to “6. Conclusions” (Page 9, Line 428).

Reviewer 2 Report

Comments and Suggestions for Authors

The manuscript is very well written. The literature review is comprehensive and interesting. Bibliographic references are relevant.

There is an important point that needs correction. The numbering of the various paragraphs is confusing. Part 3 "Disease-associated cellular events" reverts to 2 "Molecular regulation of epileptogenesis" with sub-paragraphs numbered in 4. At the end of the text the last two paragraphs (line 355 and line 432) have the same numbering and the same title.

Author Response

Response to Reviewer 2 Comments

1. Summary

Thank you very much for taking the time to review our manuscript. Please find the corresponding corrections highlighted and in track changes in the re-submitted files.

2. Questions for General Evaluation

Reviewer’s Evaluation

Response and Revisions

(Not applicable)

3. Point-by-point response to Comments and Suggestions for Authors

Comments 1: The manuscript is very well written. The literature review is comprehensive and interesting. Bibliographic references are relevant.

There is an important point that needs correction. The numbering of the various paragraphs is confusing. Part 3 "Disease-associated cellular events" reverts to 2 "Molecular regulation of epileptogenesis" with sub-paragraphs numbered in 4. At the end of the text the last two paragraphs (line 355 and line 432) have the same numbering and the same title.

Response 1: Thank you very much for the positive evaluation of our manuscript and pointing out important corrections needed in the section labeling. We have corrected the title of section 4 to “4. Molecular regulation of epileptogenesis” (Page 4, Line 195) and of section 6 to “6. Conclusions” (Page 9, Line 428).